The expression and clinical significance of serine hydroxymethyltransferase2 in gastric cancer

Shan Yiming
Liu Dongdong
Li Yingze
Wu Chu
Ye Yanwei yeyanwei66@163.com
Department of Gastrointestinal Surgery, The First Affiliated Hospital of Zhengzhou University , Zhengzhou, Henan , China
Zhang Xin
Electronic publication date: 2024 Jan 4
Publication date: 2024
Volume: 12
Electronic Location ID: e16594
Received 2023 Apr 21; Accepted 2023 Nov 14
Copyright: © 2024 Shan et al.
Copyright year: 2024
Copyright holder: Shan et al.
License: This is an open access article distributed under the terms of the Creative Commons Attribution License, which permits unrestricted use, distribution, reproduction and adaptation in any medium and for any purpose provided that it is properly attributed. For attribution, the original author(s), title, publication source (PeerJ) and either DOI or URL of the article must be cited.
License URL: https://creativecommons.org/licenses/by/4.0/

Keywords: SHMT2, Gastric cancer, Prognosis, Biomarker

Funding: National Natural Science Foundation of China 81201955 Key Scientific Research Projects of Universities in Henan Province 19A320080, 19B320010 and 23A320027 Medical Science and Technology Research Project of Henan Province SBGJ2018010 Foundation of the Department of Science and Technology of Henan Province 192102310384 Medical Science and Technology Research Project of Henan Province SBGJ202302050 This study was supported by the National Natural Science Foundation of China [Grant No. 81201955], the Key Scientific Research Projects of Universities in Henan Province [Grant No. 19A320080, No. 19B320010, No. 23A320027], the Medical Science and Technology Research Project of Henan Province [Grant No. SBGJ2018010] and the Foundation of the Department of Science and Technology of Henan Province [Grant No. 192102310384]. The Medical Science and Technology Research Project of Henan Province [Grant No. SBGJ202302050] supported the APC. The funders had no role in study design, data collection and analysis, decision to publish, or preparation of the manuscript.

==============================
Background

Gastric cancer (GC) is one of the most common malignant tumours in the digestive system. Serine hydroxymethyltransferase 2 (SHMT2) is one of the key enzymes associated with serine metabolism. However, the prognostic role of SHMT2 in GC carcinogenesis has yet to be studied.

Methods

The expression of SHMT2 in human tumors and normal tissues was detected by the Assistant for Clinical Bioinformatics and Immunohistochemistry (IHC). The relationship of the expression of SHMT2 with clinical characteristics and survival data was analysed by the chi-square test, survival analysis and online databases. Finally, the correlation between SHMT2 expression and associated signalling channels, and molecules was analysed by online databases.

Results

SHMT2 was strongly expressed in numerous human cancers. The expression rate of SHMT2 was 56.44% in GC (P = 0.018). The survival analysis indicated that patients with high expression of SHMT2 had the worse overall survival (OS; log-rank P = 0.007). The expression of SHMT2 was correlated with tumour size (P = 0.034) and, TNM stage (P = 0.042). In particular, SHMT2, vessel invasion and M stage were independent factors for OS in GC (P = 0.044, P < 0.001, P < 0.001). The SHMT2 gene was substantially correlated with cell signalling pathways.

Conclusions

SHMT2 is highly expressed in GC and is associated with a poor prognosis. The exploration of its mechanism may be related to tumour proliferation, DNA repair and replication. SHMT2 is an independent prognostic risk factor and a potential biomarker for the diagnosis and treatment of GC.

Introduction

Gastric cancer (GC) is one of the most common malignant tumours in the digestive system with high mortality globally. Research has shown that the 5 year survival rate of advanced gastric cancer is less than 10% (Zeng & Jin, 2021). Most patients were already in an advanced stage when they were diagnosed and had a poor prognosis (Chen et al., 2021). At present, targeted therapy and targeted treatment are the focus of tumour research. A previous report showed that trastuzumab could be used for patients with Her-2 gene overexpression in combination with chemotherapy (Meric-Bernstam et al., 2019). Apatinib and pembrolizumab have also shown effectiveness in the treatment of advanced GC (Kawazoe et al., 2020; Yang et al., 2022). However, their clinical benefits are so limited that the overall effect is not evident. Therefore, it is of great significance to explore new targets and effective treatments for advanced GC. In our research, we previously focused on Fibroblast Growth Factor 4 (FGFR4).

We found that the expression of the SHMT2 protein decreased when the expression of FGFR4 was knocked down; the proliferation experiment showed that the proliferation ability of SHMT2 decreased when FGFR4 was knocked down. Therefore, we chose SHMT2 for subsequent studies. SHMT2, a key enzyme of serine metabolism, has also been studied (Xie & Pei, 2021). SHMT2 plays an important role in serine metabolism, which can catalyse the transformation from serine to glycine and produce activated one-carbon units to support the synthesis of S-adenosylmethionine (SAM) and glutathione (Giardina et al., 2015; Kalhan & Hanson, 2012; Labuschagne et al., 2014). The deletion of SHMT2 expression will not be able to maintain the initiation of mitochondria-encoded protein translation (Minton et al., 2018). In a study of tumour-bound energy metabolism, SHMT2 was linked to HIF-1α which indicated that serine metabolism regulated the redox response in the anoxic phase (Ye et al., 2014). SHMT2 may also limit pyruvate kinase 2 (PKM2) and reduce oxygen consumption to accommodate the microenvironment in tumour cells (Kim et al., 2015). SHMT2 mRNA and protein are strongly expressed in colorectal cancer and gliomas and are associated with poor prognosis (Cui et al., 2022; Wu et al., 2017). The mechanism may be due to the influence of tumour proliferation and metastasis through the epithelial-mesenchymal transformation pathway and the PI3K/Akt pathway (Chen et al., 2022; Clark et al., 2022).

This article aimed to explore the relationship between the expression of SHMT2 and the clinical characteristics and prognosis of patients. In addition, the possible mechanism by which SHMT2 affects the prognosis of patients was preliminarily explored by bioinformatics analysis.

Materials and Methods

Differential expression of SHMT2 in multiple human tumours

Pancancer analysis was conducted on the expression of SHMT2 mRNA in a variety of human tumour tissues and precancerous tissues by using the UALCAN database (http://ualcan.path.uab.edu), which is a comprehensive, user-friendly and interactive web resource for analysing cancer OMICS data (Chandrashekar et al., 2022).

SHMT2 gene expression differences

First, differential expression of the SHMT2 gene was carried out from the module of TCGA and GTEX modules of the Assistant for Clinical Bioinformatics (http://www.aclbi.com) (Izzi, Davis & Naba, 2020). In addition, the difference in gene expression was analysed by the UALCAN database.

Survival and clinical data correlation analysis in GC

Prognostic analysis was implemented through the Kaplan-Meier plotter online tool (http://kmplot.com/analysis) and the 214096_s_at data set was selected to analyse and draw the survival curves of three indices including overall survival rate (OS), first progression survival time (FP) and second progression survival time (PPS).The Kaplan-Meier plotter is able to assess the correlation between gene expression and survival in multiple tumour samples. Sources for databases include TCGA, GEO and EGA (Lanczky & Gyorffy, 2021).

Immunohistochemical staining and scoring

Human gastric cancer tissue samples were purchased from Shanghai Outdo Biotech Company (No. HStma180su16), in which 101 cancer tissues and 80 normal tissues had complete clinical case data. Since the clinical cases had been screened, all the original data generated from the immunohistochemistry experiment using the microarray will be included in the scope of this statistical analysis. The paraffin-embedded tissue microarray was dewaxed and hydrated by fresh dimethylbenzene and ethanol. The antigen was repaired by the microwave repair method using 0.1 M sodium citrate buffer. After blocking the antigen, the tissues were incubated with anti-SHMT2 (#33443s, CST, 1:100) at 4 °C overnight. The sections were incubated with secondary antibodies for 15 min. The experimental data were interpreted by two experienced pathologists using the integral system. The staining intensity score was as follows: 0-negative, 1-weakly positive, 2-positive and 3-strongly positive; The score of the positive staining rate was as follows: 1-(0 ~ 25%), 2-(26 ~ 50%), 3-(51 ~ 75%), 4-(76 ~ 100%). The total score was the product of the staining intensity score and staining positive rate (Guo et al., 2021). A score <6 was defined as low SHMT2 expression, and a score ≥6 was defined as high SHMT2 expression.

Correlation analysis between SHMT2 and Ki-67, TP53, CDH1, and PROM1 in GC

The microarray also included the expression information of Ki-67, TP53, CDH1, and PROM1. The correlation analysis was performed by the correlation model of GEPIA (http://gepia.cancer-pku.cn). The Pearson test method was used to analyse the paired gene expression correlation. GEPIA is an interactive web for analysing the RNA sequencing expression data that includes single gene analysis, cancer type analysis and multiple gene analysis plates and can be used for the analysis of differential gene expression and survival analysis (Tang et al., 2017).

Relationship between SHMT2 gene and numerous pathways, MSI, and TMB

The correlation between the SHMT2 gene and pathway scores was analysed by Spearman correlation. All the analysis methods were implemented by the Assistant for Clinical Bioinformatics.

Cell lines and cell culture

The human cell lines GES-1, AGS, MGC-803, HGC-27, and MKN-45 were obtained from the Chinese Academy of Sciences, Science Bank of the Typical Culture Collection (Shanghai, China). The cell lines were cultured in DMEM (Biological Industries, Beit Haemek, Israel) supplemented with 10% foetal bovine serum (FBS, Gibco, Waltham, MA, USA), 100 U/ml penicillin and 0.1 mg/ml streptomycin (#P1400, Solarbio, Beijing, China), 37 °C, 5% CO2 incubator in a closed-culture environment.

Western blot

Total cell lysates were extracted with RIPA buffer (#R0010, Solarbio, Beijing, China). The protein samples were separated by sodium dodecyl sulfate polyacrylamide gel electrophoresis (SDS-PAGE)and then transferred to polyvinylidene difluoride (PVDF) membranes. After blocking for 1 h, the PVDF membranes was infiltrated at four degrees overnight with anti-SHMT2 (1:1,000, #33443S, CST, San Antonio, TX, USA) and anti-beta actin (1:2,500, 66009-1-Ig, Proteintech, Wuhan, China). The PVDF membrane was washed with TBST three times and incubated in a diluent containing goat anti-mouse or goat anti-rabbit IgG (1:10,000, RS23710 and RS23720, Immunoway, USA) at room temperature for 1 h. Next, the membrane was detected by a near-infrared imaging system (Image-Quant LAS 3000, General Electric Co., Fairfield, CT, Boston, MA, USA). The relative protein expression was analysed with ImageJ software.

Statistical analysis

SPSS statistics 20.0, and GraphPad Prism 8 were used for statistical analysis and plotting. The chi-square test was used to analyse the difference in SHMT2 expression in tumour and precancerous tissue and the correlation of SHMT2 expression with clinical indices. The correlation between SHMT2 expression and prognosis was analysed by Kaplan-Meier survival analysis. The variables with statistical significance in univariate analysis were analysed by Cox multivariate regression analysis. The difference in quantitative data between the two groups was tested by Student’s t test. P < 0.05 was considered statistically significant.

Ethics approval statement

The study was approved by the ethics committee of Shanghai Outdo Biotech Co. Ltd. (SHXC2021YF01) and observed the Declaration of Helsinki.

Results

The expression of SHMT2 is upregulated in many tumours including GC

First, we analysed the difference in SHMT2 expression levels between human tumours and normal tissue through the UALCAN database. The results showed that compared with normal tissues, SHMT2 is upregulated in many human tumours including GC (Fig. 1A). Subsequently, the difference in SHMT2 expression between CG and precancerous tissues was detected by using the TCGA and UALCAN databases. The results showed that SHMT2 was elevated in GC compared to normal tissues, (P < 0.001; Figs. 1B and 1C). In addition, WB results showed that the expression of SHMT2 protein in AGS (P = 0.013), MGC-803 (P = 0.001), HGC-27 (P = 0.001), and MKN-45 (P = 0.03) cells increased, compared with that in GES-1 cells (Fig. 1D).

Figure 1 The expression of SHMT2 in tumor and normal tissues or cell lines.

(A) Pan-cancerous analysis of the expression of SHMT2 in tumors. (B and C) The expression distribution of SHMT2 gene in tumor tissues and normal tissues. (D) The relative expression of SHMT2 protein in different cell lines. **P < 0.01, ***P < 0.001, ****P < 0.0001. The red represents tumor groups, the blue represents normal groups. P < 0.05 was considered statistically different. SHMT2, Serine hydroxymethyltransferase-2; STAD, Stomach adenocarcinoma.

High expression of SHMT2 is positively correlated with tumour size and TNM stage

The tissue chip contains 101 cancer tissues and 80 normal tissues. The high expression percentages of SHMT2 in cancer and normal tissues were 56.4% and 38.75%, respectively (P = 0.018). The chi-square test showed that the expression of SHMT2 was correlated with tumour size (P = 0.034) and TNM stage (P = 0.042), but not with age, sex, pathological grade, vessel invasion, or lymph node metastasis (Table 1).

Table 1 The correlation between the expression of SHMT2 and clinical features.

Clinical features	Low expression of SHMT2	High expression of SHMT2	χ 2	P	
Tissue type					
Para-carcinoma tissue	49	31	5.589	0.018	
Cancerous tissue	44	57			
Age					
<60	14	27	0.008	0.929	
≥60	21	39			
Gender					
Male	27	15	0.026	0.871	
Female	37	22			
Tumor size					
<5cm	23	17	4.493	0.034	
≥5cm	22	39			
Pathological grade					
I ~ II	7	35	0.037	0.848	
III ~ IV	9	50			
Vessel invasion					
No	34	49	1.281	0.258	
Yes	10	8			
Lymph node metastasis					
No	14	27	0.653	0.419	
Yes	16	44			
T stage					
T1 ~ T2	5	35	0.103	0.748	
T3 ~ T4	9	52			
M stage					
M0	41	3	0.088	0.767	
M1	51	6			
TNM stage					
I ~ II	21	16	4.133	0.042	
III ~ IV	23	41			
Notes:

P < 0.05 was indicated a statistically significant difference.

SHMT2: serine hydroxymethyltransferase2.

SHMT2, vessel invasion and M stage are independent factors for GC

Univariate analysis showed that SHMT2 (HR = 1.910 (1.180–3.090), P = 0.008), tumour size (HR = 2.038 (1.259–3.298), P = 0.004), pathological grade (HR = 2.363 (1.082–5.162), P = 0.031), vessel invasion (HR = 4.164 (2.366–7.327), P < 0.001), T stage (HR = 3.320 (1.209–9.116), P = 0.020), M stage (HR = 5.094 (2.370–10.950), P < 0.001), TNM stage (HR = 2.379 (1.392–4.067), P = 0.002) were prognostic factors in patients with gastric cancer. The above factors were included in the multivariate Cox regression model for analysis (Table 2). SHMT2 was considered an independent prognostic risk factor (HR = 1.683 (1.014–2.792), P = 0.044). In addition, vessel invasion and M stage were also considered (HR = 3.777 (2.105–6.780), P < 0.001; HR = 4.196 (1.890–9.315), P < 0.001).

Table 2 COX regression analysis.

Factors	Univariate analysis	Multivariate analysis	
HR	CI	P	HR	CI	P	
Gender	1.286	[0.803–2.059]	0.296	–	–	–	
Age	1.455	[0.859–2.465]	0.183	–	–	–	
Tumor size	2.038	[1.259–3.298]	0.004	1.314	[0.774–2.232]	0.312	
Pathological grade	2.363	[1.082–5.162]	0.031	2.076	[0.922–4.674]	0.078	
Vessel invasion	4.164	[2.366–7.327]	<0.001	3.777	[2.105–6.780]	<0.001	
Lymph node metastasis	1.548	[0.886–2.705]	0.125	–	–	–	
T stage	3.320	[1.209–9.116]	0.020	1.922	[0.668–5.526]	0.226	
M stage	5.094	[2.370–10.950]	<0.001	4.196	[1.890–9.315]	<0.001	
TNM stage	2.379	[1.392–4.067]	0.002	1.289	[0.700–2.375]	0.416	
SHMT2	1.910	[1.180–3.090]	0.008	1.683	[1.014–2.792]	0.044	
Notes:

P < 0.05 was indicated a statistically significant difference.

HR, Hazard Ratio; CI, Confidence Interval.

High expression of SHMT2, T stage, M stage, pathological grade, tumour size, vessel invasion and TNM stage are associated with poor prognosis

To clarify the effect of high expression of SHMT2 on the OS and PPS of patients with GC, a dataset (214096_s_at) was selected by Kaplan-Meier plotter. Compared with the low expression group, the OS, FP and PPS of patients with high expression of SHMT2 were significantly lower with the median survival times of 40 months and 25.9 months, 24.53 months and 13.2 months, 11.2 months and 8.1 months, respectively (Fig. 2B; HR = 1.27 (1.05–1.53), P = 0.012; HR = 1.31 (1.07–1.6), P = 0.0087; HR = 1.3 (1–1.68), P = 0.047).

Figure 2 Relationship between the expression of SHMT2 and prognosis in GC.

(A) The positive expression image of SHMT2 in normal and tumor tissues under 40X and 100X visual fields, respectively. (B) The correlation between the expression of SHMT2 and prognosis including OS, FP, PPS. (C–I) Kaplan-Meier survival curve including TNM stage, the expression of SHMT2, T stage, M stage, pathological grade, tumor size and vessel invasion. P < 0.05 was considered statistically different. OS, Overall survival; FP, First progression; PPS, Post progression survival; GC, gastric cancer.

The results of IHC staining revealed that the high expression rates of SHMT2 in precancerous tissues and cancer tissues were 38.75% and 56.44%, respectively (Fig. 2A). Then, Kaplan -Meier survival analysis was performed in combination with the clinical data of the patients. The results indicated that factors such as TNM stage (χ2 = 10.810, log-rank P = 0.001), high expression of SHMT2 (χ2 = 7.279, log-rank P = 0.007), T stage (χ2 = 6.181, log-rank P = 0.019), M stage (χ2 = 21.903, log-rank P < 0.001), pathological grade (χ2 = 5.006, log-rank P = 0.025), tumour size (χ2 = 8.876, log-rank P = 0.005), and vessel invasion (χ2 = 29.205, log-rank P < 0.001) were associated with poor prognosis (Figs. 2C–2I).

SHMT2 is closely related to MKI67, TP53, CDH1, and PROM1

Furthermore, we analysed the correlation between SHMT2 expression and TP53, MKI67, VEGFR, PROM1, and E-cadherin in GC. The results showed significant differences in the expression of TP53, VEGFR, PROM1 and SHMT2 expression (P < 0.05, Table 3). In addition, the correlation between SHMT2 and the above indicators was analysed through GEPIA. The results showed that the expression of SHMT2 was significantly correlated with MKI67, TP53, CDH1, and PROM1 (Figs. 3A–3D; P < 0.001, R = 0.6; P < 0.001, R = 0.49; P < 0.001, R = 0.38; P < 0.001, R = 0.15).

Table 3 Correlation between SHMT2 expression and molecules.

Molecules	Low expression
of SHMT2	High expression of SHMT2	χ 2	P	
P53					
Low expression	26	17	7.820	0.005	
High expression	15	33			
Ki-67					
Low expression	10	29	0.115	0.735	
High expression	15	37			
VEGFR					
Low expression	23	16	4.514	0.034	
High expression	19	33			
CD133					
Low expression	22	17	4.297	0.038	
High expression	18	34			
E-Cadherin					
Low expression	20	19	0.445	0.505	
High expression	23	29			
Notes:

Direction:10 cases had lost in the process of clinical follow-up and was not included in the statistical scope.

P < 0.05 was indicated a statistically significant difference.

SHMT2, serine hydroxymethyltransferase2; VEGFR, vascular endothelial growth factor receptor.

Figure 3 Correlation between SHMT2 and MKI67, TP53, CDH1, PROM1.

(A–D) The abscissa and ordinate represent the number of genes corresponding to each one million transcripts in the sample. The value on the top represents the correlation P value and correlation coefficient. P < 0.05 was considered statistically different.

The SHMT2 gene is involved in multiple signalling pathways

To further demonstrate that SHMT2 may promote the occurrence and development of GC and affect the prognosis of patients via certain pathways, we searched the relevant signalling pathways involved in the SHMT2 by Assistant for Clinical Bioinformatics. The results showed that the expression of the SHMT2 gene was closely related to tumour proliferation, DNA repair, the G2M checkpoint, MYC-targeted genes and DNA replication (Figs. 4A–4E; P = 2.85e−32, CI [0.48–0.63]; P = 8.19e−39, CI [0.53–0.67]; P = 2.7e−39, CI [0.54–0.67]; P = 8.29e−45, CI [0.58–0.70]; P = 6.76e−32 CI [0.54–0.67]).

Figure 4 The correlations between individual gene and pathway score was analyzed.

(A–E) The abscissa represents the distribution of the gene expression and the ordinate represents the distribution of the pathway score. The density curve represents the trend in the distribution of pathway immune score, the upper density curve represents the trend in the distribution of the gene expression. The value on the top represents the correlation P value and correlation coefficient. The size of the dots represents the size of the correlation coefficient. The color represents the significance of P value, the bluer the color, the smaller P value. P < 0.05 was considered statistically different.

Discussion

Currently, surgical resection combined with adjuvant chemotherapy remains the primary treatment for GC patients. It is necessary to explore effective treatment methods to improve the survival time of patients with advanced GC. SHMT2 is mainly expressed in mitochondria which can convert serine into glycine and one carbon unit and then participate in nucleotide biosynthesis, DNA methylation reactions and oxidation-reduction defence (Engel et al., 2020). A study showed that SHMT2 also played a role in the development and metastasis of breast cancer, possibly through circRNA mediated miR-149-5P overexpression of SHMT2 (Qi et al., 2020). In addition, the human oestrogen receptor-related receptor α could indirectly activate SHMT2 expression and then increase the resistance of breast cancer to lapatinib (Li et al., 2020). Zhang et al. (2022) found that SHMT2 was upregulated in oral squamous cell carcinoma and that strong expression was linked to poor prognosis. More importantly, the role of SHMT2 has been reported in glioma (Kim et al., 2015), lymphoma (Wilke et al., 2022) and bladder cancer (Zhang & Yang, 2021).

Our results showed that the expression of SHMT2 was regulated at a senior level within the GC. Compared with normal gastric cells, the results of Western blotting confirmed higher expression of SHMT2 in tumour cell lines. The correlation between the expression of SHMT2 and the prognosis of patients was further analysed. The laboratory results are consistent with the bioinformatics analysis results in terms of the impact of high SHMT2 expression on prognosis, which indicated that GC patients with high SHMT2 expression tended to have poor prognosis. Furthermore, some studies have shown that there is a significant difference between SHMT2 expression and distant metastasis, but this phenomenon was not found in this study, which may be related to few samples, and subsequent experiments may explore more accurate conclusions on the basis of increasing samples. Simultaneously, multivariate Cox regression analysis showed that SHMT2 was an independent risk factor for the prognosis of GC, which was consistent with the results of previous studies (Ning et al., 2018; Wu et al., 2020). The above results were mutually confirmed, which further illustrated that SHMT2 could affect the occurrence and development of GC and the prognosis of patients. SHMT2 may be a biomarker for targeted treatment and prognosis assessment of GC patients.

Moreover, we also analysed the expression of P53, Ki-67, VEGFR, CD133, and E-cadherin in GC, and the results showed that there was a statistically significant difference in the expression of SHMT2, P53, VEGFR, and CD133. These indicators may interact with each other and jointly influence the development of GC. Furthermore, the correlation between SHMT2 and the above genes was verified by the GEPIA database, and the results showed that SHMT2 has significant gene correlation with MKI67, TP53, CDH1, and PROM1. This means that SHMT2 may also play a role in tumour growth and metabolism.

In addition, the relationship between the SHMT2 gene and other signalling pathways was explored through the TCGA database and correlation analysis was performed. The results showed a significant correlation between the SHMT2 gene and tumour proliferation, DNA repair and replication, cellular cycle, and MYC genes, which suggested that SHMT2 might be involved in the appearance and development of tumours in a variety of processes. However, the mechanism of tumour progression mediated by SHMT2 is still poorly explored.

Although this study clarified the effect of SHMT2 on the prognosis of patients with GC from several aspects, there are still many limitations. First, we only discussed the expression of SHMT2 at the level of tissue and protein, but we did not verify the function of SHMT2 in vivo and vitro. In addition, the tissue microarray contains 101 paraffin sections, and the conclusion may be biased. In exploring which signalling pathway SHMT2 affects the occurrence and development of GC, we only conducted database analysis without further validation at the molecular level. The above limitations can be further improved in future research.

Conclusion

In conclusion, our results showed that SHMT2 was significantly upregulated in GC and that the expression of SHMT2 was related to TNM stage and tumour size. In addition, the high expression of SHMT2, advanced T stage, M stage, pathological grade, tumor size, vessel invasion and TNM stage predicted the poor prognosis with GC patients. SHMT2 expression, vessel invasion and M stage were independent prognostic factors for patients of GC. SHMT2 may be a potential biomarker to evaluate the prognosis of GC and is expected to become a new target for targeted therapy.

Supplemental Information

Supplemental Information 1 original data.

Click here for additional data file.

Supplemental Information 2 Follow-up data.

Click here for additional data file.

Supplemental Information 3 Tissue assay information.

Click here for additional data file.

Additional Information and Declarations

Competing Interests

Author Contributions

Human Ethics

Data Availability

The authors declare that they have no competing interests.

Yiming Shan performed the experiments, prepared figures and/or tables, and approved the final draft.

Dongdong Liu performed the experiments, prepared figures and/or tables, and approved the final draft.

Yingze Li analyzed the data, authored or reviewed drafts of the article, and approved the final draft.

Chu Wu analyzed the data, prepared figures and/or tables, and approved the final draft.

Yanwei Ye conceived and designed the experiments, authored or reviewed drafts of the article, and approved the final draft.

The following information was supplied relating to ethical approvals (i.e., approving body and any reference numbers):

The study was approved by the ethics committee of Shanghai Outdo Biotech Co. Ltd. (SHXC2021YF01).

The following information was supplied regarding data availability:

The raw data, including the full-length uncropped gel photos, are available in the Supplemental Files.

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
