# Peer review of "The expression and clinical significance of serine hydroxymethyltransferase2 in gastric cancer"

_PeerJ, doi:10.7717/peerj.16594_

## Round 0.1 · original submission · Major Revisions

Please revise the manuscript carefully and answer the reviewers' questions.

Reviewer 1 ·

Basic reporting

Line 63: "Therefore, it plays an important role in the exploration of new and effective targets for advanced GC" - what does "it" refer to here? Moreover, "targets" cannot be "effective". May be better to revise it into something like "Therefore, it is important to explore the new targets and effective treatment for advanced GC".

In Introduction, there seems to be a lack of a link between the first and second paragraphs - the reason for exploring SHMT2 in gastric cancer seems to come from nowhere. It would be better if more introduction could be added or the contents could be re-ogranized to bridge GC and SHMT2.

Line 106: What was the secondary antibody and what was the concentration?

Line 140: How were the quantity and quality of the extracted RNA measured?

Experimental design

no comment

Validity of the findings

Figure 1D: β-actin is supposed to be a loading control. However, from this figure, the amounts of β-actin appear inconsistent across wells. In fact, for all five lanes, the lighter SHMT2 signal, the stronger β-actin signal. Is there any explanation for that?

Figure 1E: The mRNA levels of SHMT2 in AGS and MGC-803 were not significantly different from that from GES-1, while Western Blot showed that their protein levels were significantly different from that from GES-1. Moreover, the SHMT2 mRNA levels in HGC-27 and MKN-45 were similar, while their protein levels were significantly different. This discrepancy is concerning. The authors mentioned that it could be due to posttranslational modification of SHMT2 mRNA in Line 277. However,
(1) posttranslational modification is on proteins, not on mRNA;
(2) there should be added reference(s) showing that SHMT2 may have posttranslational modification, for example, https://www.ncbi.nlm.nih.gov/pmc/articles/PMC8408114/
(3) even if SHMT2 does have posttranslational modification, why would it be so different among different tumor cell lines? Either more discussion should be provided, or more experiment evidence should be provided to show that the data shown are valid.

·

Basic reporting

The manuscript identified SHMT2 highly expressed in gastric cancer, which was associated with tumor proliferation, DNA repair and replication.
1. What’s the five year survive rate for gastric cancer? The authors should list it in the Introduction section.
2. What’s the difference between cancer tissues and normal tissues? Why the number of cancer tissues and normal tissues is not same?

Experimental design

The authors didn’t verify the function of SHMT2 in vivo and vitro. What’s the future procedures and plans for the identification of the function of SHMT2 in vivo and vitro.

Validity of the findings

In Fig. 1A, what’s the identification of different tumors? Do there have any references involved in these tumors?

---

## Round 0.2 · Major Revisions

The author must revise the manuscript carefully. Must be polished in English and provided with proof of editing.

**Language Note:** The Academic Editor has identified that the English language must be improved. PeerJ can provide language editing services - please contact us at [email protected] for pricing (be sure to provide your manuscript number and title). Alternatively, you should make your own arrangements to improve the language quality and provide details in your response letter. – PeerJ Staff

Reviewer 1 ·

Basic reporting

The whole manuscript needs extensive revision in English writing. Throughout the manuscript, there are typos or grammatical errors which can hinder the understanding of the scientific concepts, or even may prevent some readers from continuing reading. Below are just some examples.


Line 58: It's confusing to say "patients were included in advanced stage" - do you mean "patients were at the advanced stage"?

Line 60: "Previous report shown" should be "Previous report has shown".

Line 64: "their clinical benefits are so limited that the overall effect is not evident" this sentence is confusing, especially because you just mentioned that they had shown their effectiveness in the treatment of advanced GC.

Line 66: "previous" is redundant.

Line 67: "would decreased" should be "decreased".

Line 68: "shown" should be "showed".

Figure 1: It would be better if it could be consistent that on the graphs, Tumor group is on the left and Normal group is on the right; or consistent that it's vice versa in all graphs.

Experimental design

no comment

Validity of the findings

no comment

Additional comments

no comment

·

Basic reporting

No comment

Experimental design

1. In line 67-68, the authors found that the expression of SHMT2 protein would decrease when the expression of FGFR4 was knocked down. What’s the mechanism of the phenomena?
2. In Fig.1D, what’s the transcription level and protein expression of FGFR4 in different cell lines?
3. In Fig.4, how many tumor proliferation, DNA repair and
305 replication, cellular cycle, and MYC genes involved in the analysis?

Validity of the findings

1. How many types of gastric cancer? What’s the identification of advanced gastric cancer?
2. How many stages of gastric cancer? What’s the identification of advanced stage?
3. The authors have previously focused on the molecule of fibroblast growth factor 4 (FGFR4). What’s the reference of the previous research?
4. The English language should be improved to ensure that an international audience can clearly understand your text. Some examples where the language could be improved include lines 67, 70, 276.

---

## Round 0.3 · accepted · Accept

Both reviewers gave positive opinions. After reviewing the manuscript, I basically agree with the two reviewers. This manuscript meets the requirements for publication, and there is no obvious risk of publication.

Reviewer 1 ·

Basic reporting

no comment

Experimental design

no comment

Validity of the findings

no comment

Additional comments

My previous concerns/comments have been appropriately addressed.

·

Basic reporting

The article shared the raw data and professional figures.

Experimental design

The authors are well defined research questions.

Validity of the findings

Conclusion are well stated.